# High-Throughput Monoclonal Antibody Discovery from Phage Libraries: Challenging the Current Preclinical Pipeline to Keep the Pace with the Increasing mAb Demand

**DOI:** 10.3390/cancers14051325

**Published:** 2022-03-04

**Authors:** Nicola Zambrano, Guendalina Froechlich, Dejan Lazarevic, Margherita Passariello, Alfredo Nicosia, Claudia De Lorenzo, Marco J. Morelli, Emanuele Sasso

**Affiliations:** 1Dipartimento di Medicina Molecolare e Biotecnologie Mediche, Università Degli Studi di Napoli Federico II, Via Pansini 5, 80131 Napoli, Italy; froechlich@ceinge.unina.it (G.F.); margherita.passariello@unina.it (M.P.); nicosia@ceinge.unina.it (A.N.); cladelor@unina.it (C.D.L.); 2CEINGE—Biotecnologie Avanzate s.c. a.r.l., Via Gaetano Salvatore 486, 80145 Naples, Italy; 3Center for Omics Sciences Ospedale San Raffaele, Via Olgettina 58, 20132 Milano, Italy; lazarevic.dejan@hsr.it (D.L.); morelli.marco@hsr.it (M.J.M.)

**Keywords:** monoclonal antibodies, high-throughput screening, next-generation sequencing, third-generation sequencing, target-unrelated, selection-unrelated, selection-related, mAbs production

## Abstract

**Simple Summary:**

Monoclonal antibodies are increasingly used for a broad range of diseases. Rising demand must face with time time-consuming and laborious processes to isolate novel monoclonal antibodies. Next-generation sequencing coupled to phage display provides timely and sustainable high throughput selection strategy to rapidly access novel target. Here, we describe the current NGS-guided strategies to identify potential binders from enriched sub-libraires by applying a user-friendly informatic pipeline to identify and discard false positive clones. Rescue step and strategies to boost mAb yield are also discussed to improve the limiting selection and screening steps.

**Abstract:**

Monoclonal antibodies are among the most powerful therapeutics in modern medicine. Since the approval of the first therapeutic antibody in 1986, monoclonal antibodies keep holding great expectations for application in a range of clinical indications, highlighting the need to provide timely and sustainable access to powerful screening options. However, their application in the past has been limited by time-consuming and expensive steps of discovery and production. The screening of antibody repertoires is a laborious step; however, the implementation of next-generation sequencing-guided screening of single-chain antibody fragments has now largely overcome this issue. This review provides a detailed overview of the current strategies for the identification of monoclonal antibodies from phage display-based libraries. We also discuss the challenges and the possible solutions to improve the limiting selection and screening steps, in order to keep pace with the increasing demand for monoclonal antibodies.

## 1. Introduction

In the current era of targeted therapy, monoclonal antibodies (mAbs) represent a powerful therapeutic tool to treat a wide range of diseases. The first monoclonal antibody was generated in 1975, and the first one was approved for clinical use in 1986 [1]; since then, this class of pharmaceuticals has demonstrated unprecedented levels of clinical and commercial success. With the approval by the FDA in April 2021 of GlaxoSmithKline’s PD-1/PD-L1 blocker Jemperli (dostarlimab), the 100th monoclonal antibody is now available for clinical use in the USA. Interestingly, half of these 100 monoclonal antibodies have been approved during the last five years (2016–2021), underlining the rapid expansion for the availability of this class of therapeutics in clinical practice. The global annual antibody market size in 2021 has approached 150 billion dollars and is estimated to duplicate by the end of 2026 with a growth rate (CAGR) of about 11% [2,3]. The successful development of mAbs for the treatment of chronic diseases (e.g., rheumatoid arthritis), the renaissance of inhibitory mAbs for the early treatment and diagnosis of infectious diseases due to the COVID-19 pandemic, together with the patent expiry of several successful antibodies (e.g., adalimumab, infliximab, rituximab, trastuzumab, ipilimumab) are boosting the growth of the antibody market [4].

For years, due to challenging screening technologies and the high manufacturing costs, mAbs have been used in clinical practice only for stringent uses. Indeed, more than half of the currently FDA approved mAbs are for the oncological therapeutic area where, if available, cheaper therapeutic options are preferred. Accordingly, the high costs have been the main obstacle for patients to access antibodies [5], particularly for infectious diseases where, despite mAbs being demonstrated as very effective, more affordable therapeutics are available (e.g., vaccines or hyperimmune sera) [6,7,8,9,10,11]. However, high-throughput technologies are shifting the cost-efficacy balance, allowing to widen mAb treatment options to many therapeutic areas for which monoclonal antibodies are poised to play an important role (e.g., infectious diseases, rheumatology, neurology). The most common technologies to obtain monoclonal antibodies are based on mouse hybridoma, phage display and single B cell screening technologies. The hybridoma technology takes advantage of the host’s natural ability to generate B cells able to produce functional, specific and high-affinity antibodies against a foreign antigen [12]. However, there are some significant issues limiting the use of the hybridoma technology: (i) the length of the process (3–4 months); (ii) the animal origin of antibodies; and (iii) the low efficiency or lack of feasibility in the selection of antibodies against highly conserved proteins between human and mouse or against toxic agents (e.g., for toxicology and antivenom research). To overcome immunogenicity of mouse antibodies in clinical settings, due to human anti-mouse antibody (HAMA) responses, many efforts were dedicated to graft murine variable regions or complementary determining regions (CDRs) into human antibody backbones to generate chimeric and humanized mAbs, respectively [13]. More recently, transgenic mice expressing human antibodies and single human B cell techniques have been developed to overcome the HAMA issue and the challenge of CDRs grafting [14,15]. The first approved chimeric and humanized antibodies were the anti-GPIIb/IIIa Abciximab in 1994, and anti-IL-2 daclizumab in 1997, respectively [16,17,18,19]. The application of mAbs to preclinical and clinical studies rapidly increased in the 90s, thanks to the application of phage display technology to antibody libraries, allowing to obtain fully human monoclonal antibodies against virtually any target [20,21,22]. Briefly, in the phage display technology, foreign sequences (peptides or antibody fragments) are fused to one of the coat proteins (pIII or pVIII) of the filamentous bacteriophage M13, to display them as surface molecules. To generate phage displayed antibody repertoires, the single-chain variable fragment (scFv) format was developed by fusing variable regions of heavy (VH) and light (VL) chains with the insertion of a flexible linker. The source of variable heavy and light regions is usually the entire repertoire of cDNAs recovered by RT-PCR from healthy donors’ human B cells (naïve library). The VH and VL are thus randomly assembled to expand clonal assortment before cloning them into phagemid vectors. An error-prone PCR step is often used to further widen repertoire diversity [23]. Beyond phages, several different display platforms are currently available, including, for instance, ribosomal, bacterial, yeast and mammalian displays [24,25]. Due to the absence of in vivo affinity maturation, monoclonal antibodies generated by display technologies from libraries of healthy donors have often a moderate affinity, compared to those from immunized libraries (e.g., hybridoma). To overcome this limitation, in vitro affinity maturation of CDRs or in silico modeling can be applied once an antibody fragment of interest has been identified [26,27]. As an alternative to the naïve libraries, hyperimmune libraries can be generated by recovering variable regions from patients undergoing antigen stimulation, as it was recently done for the identification of high-affinity mAbs against the Spike protein of SARS-CoV-2 [28]. Beyond the scFv format, different layouts of antibody fragments have also been used to construct phage display libraries, including the antigen-binding fragment (Fab) or the single domain antibody (sdAb) [29,30,31,32]. Antibody libraries are surveyed by several affinity selection and screening cycles, each one consisting of (1) incubation of the phage repertoire with the target of interest (biopanning); (2) intensive washes; (3) elution; (4) amplification step in *E. coli*; and (5) extensive ELISA assays to identify specific binders. The molecular baits used in biopanning are usually recombinant proteins or target-expressing cells. Subtractive selection cycles are commonly included in the screening to remove clones recognizing non-target related molecules, i.e., the Fc fragment for Fc-tagged recombinant proteins, or cells not expressing the target of interest (Figure 1) [33]. Phage selection was also developed in vivo by performing selection cycles in living animals; this strategy is properly applied when the target of interest is modified upon in vitro modeling [34,35]. Adalimumab was the first human monoclonal antibody, isolated by phage display with a “guided selection”, to be approved for clinical applications in 2002 [36].

This notwithstanding, the phage antibody library selection and screening process is slow, expensive, and often limited by the number of clones that can be tested by binding assays, thus representing a bottleneck to expand the use of mAbs. Emerging technologies are resizing this hurdle, paving the way to an expanded use of this approach.

To this end, a very significant milestone in antibody discovery by phage display was the application of next-generation sequencing (NGS) to the screening procedure. The application of NGS in place of standard screening methods, requiring intensive use of ELISA (or other binding assays), has dramatically increased the throughput and decreased the working time of the phage antibody library discovery platform. The NGS-based screening proved to be beneficial for isolating potentially therapeutic antibodies for both cancer immunotherapy [33], where novel clinically relevant targets continuously emerge, and for infectious diseases, especially for emergency or preparedness applications [37,38,39]. In this review, we summarize the NGS-based technologies applied to phage display and novel approaches to speed up the identification of large numbers of candidate binders, and to improve the yields of recombinant monoclonal antibodies.

## 2. Implementation of Next Generation Sequencing to Phage Display scFv Library Screening

As mentioned before, more than half of currently approved antibodies have been developed for clinical uses in oncology [40]. Immune checkpoint inhibitors are one of the most rapidly growing classes of antibodies, with hundreds of novel investigational mAbs hitting both suppressive and activating immune circuits. Among them, mAbs targeting PD-1/PD-L1 remain over-represented in the antibody pipeline, as both single agents and in combination treatments, with many immunotherapeutic agents, including oncolytic viruses [41,42,43,44], chemotherapy and immune-radiotherapy [45]. The growing discovery of novel immune checkpoint modulators (e.g., TIM-3, VISTA) and the identification of novel mechanisms of the action of such antibodies targeting well-known axes (e.g., PD-1 and CTLA-4)—but possibly requiring different modes of binding and, therefore, different paratope sequences—underline the need to apply high-throughput screening (HTS) technologies to the mAbs discovery [46,47,48].

Since next-generation sequencing (NGS) technologies were introduced in the early 2000s, the field of life sciences has undergone a dramatic revolution [49]. The routine availability of high-throughput sequencing has progressively reduced the time and lowered the costs of drug development [10,50]. Besides the obvious applications to genomics (genome sequencing and gene expression profiling), NGS was exploited for high-throughput screening approaches of large repertoires of coding and non-coding nucleic acids (e.g., aptamer SELEX and peptides) [51].

Traditionally, Sanger sequencing is used in combination with ELISA assays to assess the identity of binder scFv clones after some rounds of selection. This analysis is limited to hundreds of clones, and it often results in repetitive isolation of the same clones that become dominant during selection cycles. In addition, clone enrichment can also reflect features not necessarily related to target affinity, including biological fitness and binding to non-target substrates (discussed in the next sections) reducing the isolation of true binders [52].

The application of NGS to antibody display technologies (such as phage display, *E. coli* display, yeast display, ribosome display) allows us to deeply characterize the library composition, exploiting the physical linkage between phenotype (peptide or scFv protein) and genotype (coding sequence) of each phage genome present in the phage pool (Figure 2) [53,54]. Based on this concept, NGS was adopted for the screening of antibody libraries to identify potential binders by monitoring the enrichment of a given sequence during the selection cycles [55]. Parallel comparative screenings of antibody libraries by traditional ELISA and NGS have demonstrated that, beyond top-ranking clones efficiently identified by both technologies, deep sequencing enables the discovery of many more specific binders among rare clones with potentially interesting features [38,56].

### 2.1. Sequencing-Guided Antibody Discovery: State-of-the-Art and Beyond

The principal drawback of high-throughput sequencing applied to antibody display libraries is related to the limited maximum read length attainable with classical NGS platforms, which is far from the average size of scFv and Fab formats (respectively 800 and 1500 bp). The most used NGS platforms in the antibody discovery field are MiSeq (Illumina, San Diego, CA, USA), Roche 454 (now discontinued) and Ion Torrent (Life Technologies) (Waltham, MA, USA), which offer a good balance between the maximum read length and acceptable error rate (about 300 bp and 0.1%, 600–800 bp and 1%, and 400 bp and 1%, respectively). In any case, the whole sequence information of an scFv is split into smaller fragments corresponding to the variable heavy (VH) and variable light (VL) regions [57,58,59,60]. Most of the sequencing strategies adopted so far have focused on sequencing only heavy chain complementary determining regions 3 (HCDR3) which is usually the one mostly contributing to binding, thereby disregarding information about other CDRs of heavy (HCDR1, HCDR2) and light (LCDR1, LCDR2, LCDR3) chain domains [57]. The paired-end reads, in part, compensate for the limited length of reads, allowing the sequencing of at least an entire variable domain (VH or VL), and improving the global error rate by reading the CDRs twice (Figure 2) [57,61,62]. Technical and bioinformatic adjustments can further improve the sequence outputs [63]. Among technical adjustments, the one implemented in Geyer’s lab was of particular interest; they applied the Kunkel mutagenesis to delete less informative framework regions between heavy and light variable domains, thus obtaining more extensive information about the scFvs’ hypervariable regions [64]. Taking into consideration the features of NGS platforms, a usual approach focuses on sequencing the entire VH by paired-ends 2 × 300 on the Illumina MiSeq platform.

The crucial steps in the NGS-guided antibody discovery are the fragment generation and library preparation. In our experience, the excision of variable fragments of interest by restriction enzymes (if useful sites are available) has to be preferred over PCR to avoid the loss of relative representativeness of clones and the potential polymerase errors that could generate artefactual sequences [61]. Indeed, although increasing the coverage allows to easily identify co-sequencing errors, the recognition of pre-sequencing polymerase errors that occurred during library preparation is a challenging task [63]. In addition, identification of truncated clones (bearing only VH, or VH with part of VL) has been reported in many studies, presumably due to their biological advantage (discussed in the next sections). To exclude these ‘contaminant’ clones from analysis, the introduction of a double step of enzymatic restriction to pull out VHs of interest only from full-length scFvs for fragment library preparation, is actually a routinely adopted, suitable option (Figure 2) [33,47]. As an alternative to enzymatic excision, a few nested ultra-high-fidelity PCR cycles can be implemented [57].

In recent years, third-generation sequencing platforms (PacBio and Oxford Nanopore Technologies) have been developed to generate ultra-long reads from single molecules without the need for PCR amplification. While the use of these real-time sequencing platforms was initially disadvantaged by low-throughput and high error rate (>10%), chemically upgraded platforms (PacBio RS II and Sequel series) as well as bioinformatics algorithms (e.g., Nanopore’s Bonito) have dramatically improved their output, allowing to obtain very long reads and a sharp reduction in the costs. The greatest breakthrough in third-generation sequencing was the development of circular *consensus* sequencing, by which a single molecule can be circularized and sequenced several times to radically reduce the error rate. These technical and bioinformatic improvements have led to obtaining ultra-long (up to 25 kilobases) and increasingly accurate (99.9% for PacBio; 97% for Nanopore) reads [49,65]. The implementation of long accurate reads and the recent introduction of sample multiplexing by barcoding renders the third-generation sequencing platforms very useful to obtain quantitative information of full-length antibody fragments (scFv, Fab) from library screenings. To date, these technologies are still very rarely applied to the screening of antibody libraries, but the recent improvements and a more widespread availability of instruments are paving the way to their more diffused application (Figure 2) [66,67,68].

### 2.2. Filtering of Potential Binders: The Funnel Approach

An effective script for NGS-guided identification of target-specific scFvs relies on the application of computational algorithms able to prioritize sequences according to the potential binding. Three crucial steps, discussed in detail in the next sub-sections, need to be implemented to identify: (1) the breakpoint selection cycle; (2) the non-valid inserts (clones bearing stop codons or frameshift indels); (3) the target-unrelated binders. (Figure 3).

#### 2.2.1. Identification of Break Point Selection Cycle

A typical scFv library is comprised of up to 10^11^–10^13^ different clones [69]. Potential binders from phage display libraries are usually selected by a number of selection cycles with the given target antigen (biopanning) followed by amplification in the *E. coli* host. Clonal diversity of the naïve library is sharply decreased during these selection cycles, as non-binder clones are counter-selected throughout biopanning [55,61]. Consequently, the sub-libraries resulting from consecutive selection cycles become composed of less and less different clones, with an increasing level of representativeness. The aim of NGS application to antibody library screening is to evaluate the trend of enrichments during the selection cycles for thousands of these scFvs, as an indicator of their selective binding. In principle, the ranking of representativeness of individual clones in a given selection cycle allows us to obtain a rough estimate of the clone binding potency to the antigen as, in principle, phage bearing scFvs with low reactivity tend to be counter-selected throughout selection cycles [55]. As expected, after several selection cycles, all the scFv binders reach a plateau of enrichment, thus flattening their relative abundance. For this reason, it is crucial to identify the earliest break point selection cycle to avoid losing the relative enrichment information.

In the last years, we applied phage display of scFv libraries to several distinct targets encompassing viral antigens and human proteins with immunomodulatory activity. Interestingly, all along with these studies, we evidenced that the second and third selection cycles were usually the most informative ones, being characterized by a sharp drop in the entropic features of the library and by the sharp enrichment of the top-ranking clone sequences [33,38,39,47,61]. The nature of the target antigen, as well as the panning and elution strategies, contribute to defining the trend of the enrichments. In the manuscript by Passariello et al. (2021), we screened by three selection cycles, a phage library of scFv against the receptor-binding domain of the Spike protein from SARS-CoV-2, comparing chemical (acidic pH) and competitive (recombinant ACE2) elution methods. Interestingly, more than half of the top-ranking clones were commonly enriched between the differentially eluted sub-libraries, but their kinetics of enrichment resulted completely differently. In particular, the competitive elution by using ACE2 recombinant protein resulted in a more gradual enrichment of top-ranking clones, compared to pH elution that flattened the relative representativeness of clones by the second selection cycle [38]. In Sasso et al. (2018) we performed a massive parallel screening of scFvs targeting several immune checkpoint modulators (LAG-3, PD-L1, PD-1, TIM-3, BTLA, TIGIT, OX40, 4-1BB, CD27 and ICOS) [33]. The screening consisted of a shared first selection cycle performed on activated human PBMCs expressing high levels of the native antigens of interest, followed by divergent and parallel biopanning cycles on Fc-fused recombinant target proteins. Despite the identical screening conditions exploited for these parallel selection cycles (reagents, elution method), the kinetic of sub-library saturation resulted dissimilarly between the different targets. Indeed, the second and the third selection cycles resulted in the breakpoint, respectively, for 3 (OX40, LAG-3, PD-L1) and 6 (PD-1, TIM3, BTLA, 4-1BB, CD27, ICOS) out of the 10 targets. Moreso, the normalized maximal relative enrichments (count per million, CPM) were highly heterogeneous, ranging over 4 orders of magnitude for the different targets. Such evidence suggests that NGS-guided screening of scFvs needs to be target-optimized and demonstrates how the definition of a standard operating procedure is challenging [33]. For these reasons, the sequencing of a single selection cycle offers only a limited picture of the entire screening process, with the risk of missing the essential snapshot. Multiplexing allows different libraries to be pooled and sequenced simultaneously during a single run on MiSeq Illumina to reduce the sequencing costs, yet still obtain informative coverage. Given this opportunity, performing a wide screening with three or more selection cycles, followed by a retrospective identification of the selection break point, could represent the optimal strategy.

#### 2.2.2. The Issue of False-Positive Clones: Target-Unrelated and Non-Valid Insert Identification

Even though the phage display technology has proven to be very effective in the discovery of monoclonal antibodies and peptides, the corresponding screening procedure is often challenged by false-positive clones that typically arise during the selection steps. These false-positive clones remain a major hurdle in library screening, as the reason for their enrichment is hard to define and, thus, to deal with. These target-unrelated false-positive clones can arise during selection steps, mainly as the result of two different biases: (1) propagation advantage (selection-unrelated); (2) binding to panning components (selection-related) (Figure 1 and Figure 3).

Propagation advantage is a selection-unrelated phenomenon. It has been extensively characterized for phage libraries as the result of the Darwinian evolution process during biopanning and repetitive amplification steps. Within an experiment, starting from the naïve phage library, the enriched sub-libraries obtained at the end of each selection cycle (negative and positive selection with the target of interest), are amplified in bacterial cells in the presence of a helper phage as preparatory to the next panning cycle. Selection-unrelated contaminant clones can arise, as well, during this intermediate amplification step due to the growth rate advantage [70]. Very often, these contaminating clones bear frameshift or non-sense mutations into the scFv coding sequence, which produces premature stop codons that impede the translation of the C-terminus protein pIII. During the amplification step, however, these clones can incorporate protein pIII supplemented by the helper phage, thus acquiring an enhanced growth rate. Moreso, clones bearing truncated forms of fusion proteins (e.g., clones bearing large in-frame deletions of scFv) can acquire a growth rate advantage due to lower interference between pIII and F *pilus* of bacteria needed for amplification [71,72,73]. This well-known issue can be at least in part circumvented by the exploitation of helper phages lacking effective infectivity domains of protein pIII to make phages infective only in the presence of full-length scFv-pIII fusion protein [74]. Deep sequencing of the whole variable heavy domains has to be preferred to a HCDR3-focused strategy, as these mutations can be equally distributed over the entire scFv. In silico translation of NGS-derived sequences allows to easily identify and discard these non-sense clones by accessible and reliable computational algorithms, as described in the next sections (Figure 3).

Identification of selection-related false-positive clones is even more challenging. To fully understand the origin of these clones, one should keep in mind that the antigen of interest is just one among the components of a selection system, including a solid-phase (e.g., plates, magnetic beads, tubes), capturing baits (e.g., Fc fragment, streptavidin, bovine serum albumin), or eventually cellular substrates expressing the antigen of interest among thousands of different molecules. The stickiness of phage particles to the scaffolds and to the macromolecular components used during the selection cycles further worsens the enrichment of those selection-related false-positive clones [75]. Negative selection steps can, of course, remove, at least in part, these non-specific phages. In the case of an antibody phage display screening performed on living cells, it is quite hard to identify an ideal substrate for negative selection, unless two background-matched, isogenic cell lines differing only in target expression are available (e.g., use of target overexpression in a target-negative cell line or knock out cell clones) [39,76,77,78]. Removal of phages reactive for molecular baits as well as solid-phase components is also challenging. Negative selection steps can be performed by incubating phage libraries with a molar excess of all the biopanning components, with the exception of the target itself (e.g., immobilized Fc on plates). Several screening strategies can be further developed to reduce this bias, including (1) masking unwanted epitopes with blocking antibodies (e.g., anti-Fc antibodies); (2) eluting target-bound phage by competition with specific soluble antigens [38,79,80]. Despite these strategies, selection-related, target-unspecific residual clones contaminate the sub-libraries and are amplified during *E. coli* infection steps. Obviously, this issue also disturbs the screening of phage display peptide libraries. The Huang’s and Smith’s labs have developed bioinformatic tools, mainly based on the target-unrelated peptides data bank (predicted and verified) to discard a priori these clones streamlining the peptide discovery process [52,81,82,83,84,85,86]. By collecting growing NGS data, we are developing a similar tool for scFv phage display libraries, described in the next sections, that would be of potential interest for the whole scientific community involved in antibody discovery.

#### 2.2.3. A Computational Strategy for Rapid Discovery of Potential scFv Binders

The aim of applying NGS to the screening of scFv phage libraries is to process the raw sequencing data into potential qualitative binding information. While direct proportionality between sequence enrichments and scFv binding helps to identify phage binders, the presence of false-positive clones strongly affects the analysis. Thus, it is essential to identify and discard contaminant sequences from NGS data output. To improve the confidence of in silico binding predictions, we propose a six-step “funnel” pipeline, implemented with an R script and starting from a standardized spreadsheet, to clean up sequencing data from false-positive clones (Figure 3), which is described below.

Step (1): A variable number of total reads is obtained for each sequenced sample from an NGS run. Thus, the raw counts for each sequence must be normalized to the total number of reads obtained for the corresponding selection cycle (e.g., counts per million, cpm). This normalization allows us to evaluate the trend of enrichment of a given sequence between selection cycles. As previously discussed, we assume that different clones can arise during selection cycles with different kinetics. For this reason, the first filter is applied to restrict the analysis to those sequences whose number increases (positive delta, ∆+) at least between two selection cycles of interest (e.g., cpm cycle3-cpm cycle2 > 1 or cpm cycle2-cpm cycle1 > 1). In this way, part of the background contaminant clones is removed from the analysis. As the typical enrichment (empirically evaluated) of a good binder clone exceeds two orders of magnitude between selection cycles, to further restrict this analysis, a more stringent threshold value can be introduced as ∆+ (e.g., Log(10)) (Figure 3) [33,38,47,61].

Steps (2–4): As previously discussed, the screening-unrelated clones can arise during the selection steps due to the gaining of a positive fitness and faster growth rate. This phenomenon can be associated with the enrichments of clones, lacking full-length valid inserts (scFv or peptide). These non-valid inserts can correspond to clones bearing stop codons in the scFv sequences, as well as from the translation of proteins on an altered open reading frame resulting from indels (presumably less toxic or generating premature stop codons). While premature protein translation within the scFv coding sequences affects the synthesis of a functional pIII protein, the latter could be incorporated from the helper phage used during the amplification steps of the naïve library, as well as from the enriched sub-libraries in the *E. coli* host. To identify these clones, we first translate the variable heavy sequence according to the open reading frame of interest. Then, we apply a cascade of three filters to respectively discard from the analysis: (1) clones not starting with framework 1 (FR1) consensus (i.e., MA-EVQ or MA-QVQ); (2) clones bearing stop codons; (3) clones whose VH does not end with framework 4 (FR4) consensus (TVS). These parameters can be easily adapted to any scFv or nanobody libraries differing in the sequences of the framework regions (Figure 3).

Step (5): The phage display relies on the binding of scFvs to the target of interest during selection cycles. It is therefore expected that synonymous phages (differing in DNA sequences but still encoding the same scFv primary structure) share similar trends of enrichment. Although the codon usage can affect the translation efficiency and display, we actually often identified synonymous sequences among the top-ranking clones in enriched sub-libraries. To give more weight to the binding potential, the counts for synonymous clones are added together and the number of different clones contributing to the counts are tracked (Figure 3).

Step (6): The last step of the proposed pipeline identifies selection-related, target-unrelated clones. To do so, amino acidic sequences are matched to an external database containing biopanning data collected from previous NGS-guided phage display scFv screenings. This database includes sequences of variable heavy chains obtained from the screening of different scFv phage libraries and also includes information about their ranking in the different enriched sub-libraries. As hCDR3 plays a dominant role in antigen binding, this analysis can be eventually restricted to this fragment (Figure 3). Clones identified in the target-unrelated database are thus discarded to further improve the confidence of binding predictions.

Here, by retrospective analysis of our previous screenings, we show how the implementation of this script is radically improving the quality of NGS-guided discovery of scFvs from phage libraries.

In the first attempt to apply NGS to mAbs discovery, we sequenced VH pools from four biopanning cycles performed on claudin-1 (CLDN-1) expressing cells [61]. By applying steps 1 to 5 of our script, we discarded 26% of sequences, as they contained stop codons or indels (altered ORF). Not having a target-unrelated database for comparison, all the clones with valid inserts (w/o stop codon and ORF alteration) were considered as potential binders. Of those, only 18% of retrieved clones resulted to be specific for CLDN-1, as validated by ELISA assays (Figure 4).

By applying the complete pipeline implemented in Sasso et al. (2018), we obtained a snapshot of third selection cycles for nine different parallel screenings. We revealed that among the 100 top-ranking clones, the percentage of valid inserts (*w*/*o* stop codons or altered ORF) ranged from 4% (for PD-L1) to 90% (for OX40 and 4-1BB) of enriched sub-libraries. Previous screening to target CLDN-1 and this screening itself allowed us to collect enough NGS data for scanning and excluding target-unrelated clones. In this way, among the sequences with a valid insert, we reported and discarded from 30% (for CD27) to 90% (for LAG-3) of clones that were present in the target-unrelated database (Figure 4). The resulting potential binders were rescued and tested by ELISA assays, showing that about half of them were specifically bound to the target of interest [33].

By exploiting a more updated target-unrelated database, in Passariello et al. (2020) we approached the highest confidence in binding prediction to human CTLA-4. Moreover, by following the enrichment of those scFv sequences into parallel selections with murine CTLA-4, their human/mouse cross-reactivity was predicted. Indeed, all the tested clones resulted specifically for human CTLA-4 and, if predicted, cross-reactive to murine orthologue (Figure 4) [33,47]. These data underline the significance of generating an accessible tool for scanning, reporting and excluding validated and predicted target-unrelated scFvs to further improve the confidence of the binding prediction.

## 3. Rescuing and Cloning Strategies

As discussed, with the exception of third-generation sequencing platforms, the linking information between heavy and light chains of binder scFvs is lost in NGS-guided screening, due to short read length that provides information at the utmost on the entire variable heavy region. That said, molecular biology strategies can be implemented to identify the actual variable light chains paired to the VHs of interest. An additional limitation of sequencing-guided high-throughput screening of scFv libraries faces the necessity to physically recover clones of interest from the library mixture, to validate the binding by biochemical assays (e.g., ELISA). To solve both issues, starting from enriched sub-libraries, different approaches to rescue clones of interest have been developed, all exploiting the uniqueness of the heavy chain CDR3s (Figure 5).

By implementing inverse PCR, a pair of overlapping primers are designed within the HCDR3 of the clone of interest, to make several copies of the given phagemid vector from the sub-library mixture. *Dpn*1 enzyme digestion results in the elimination of the methylated and hemi-methylated library DNA template. Newly synthesized DNA is thus transformed into *E. coli*; the living host repairs the nicks and upstreams the 5′ ends of the synthetic primers included in the newly synthesized copies of the phagemid of interest (Figure 5A) [61]. Instead of overlapping oligonucleotides, a reverse 5′-phosphorylated primer in FR4 can be used, in combination with a 5′-phosphorylated forward primer in the CDR3. The latter strategy needs an intermediate ligation step before transformation, to circularize the dsDNA [87]. The limitation of both inverse PCR approaches is related to the inadequate amount of recovered DNA, as newly synthesized DNA cannot act as the template for following PCR amplification cycles. For this reason, while this approach is rapid and very effective to rescue abundantly represented clones from enriched sub-libraries, it could fail to retrieve rare scFvs, probably diluted among hemi-methylated library DNA on which *Dpn*1 has a limited activity (Figure 5A) [88]. To rescue the less represented clones, an overlapping PCR method can be adopted (Figure 5B). With this approach, the scFvs of interest are reconstituted, starting from the annealing and extension from two intermediate PCR fragments of VH and VL sharing a HCDR3 region of choice for overlap. Indeed, clone-specific forward and reverse primers within the HCDR3 are combined to primers designed in the framework regions upstream and downstream from the scFv (HFR1 and LFR4) to generate HFR1-to-HCDR3 and HCDR3-to-LFR4 fragments. The second amplification step uses these two fragments to generate the full-length scFv, thanks to the shared presence of the overlapping regions in HCDR3, via PCR extension and amplification; the latter is performed with the external HFR1 forward, and LFR4 reverse primers, to actually amplify the scFv as preparatory to the ligation step in an empty phagemid vector (Figure 5B). In our experience, although more laborious, overlapping PCR allows retrieval of rare clones with an improved sensitivity of up to two orders of magnitude, compared to single-step inverse PCR [33,55,59]. A variant of PCR-based rescue has been adopted by Bradbury’s lab to rescue clones from the yeast library of scFvs in VL-linker-VH orientation, by using a universal forward primer upstream VL with a HCDR3-specific reverse primer that also includes the HFR4 [89].

A PCR-free technology to retrieve scFvs from the library uses biotinylated probes complementary to the HCDR3 as a molecular bait to fish out scFvs (Figure 5C). Unlike the PCR-based approaches, this method uses single-stranded DNA extracted directly from phage particles and hybridized to the probe. Albeit the absence of amplification could limit the rescue capacity for rare clones, this method has been validated as more selective, avoiding the promiscuity and background of PCR-based methods [90]. In any case, Sanger sequencing of the recovered clones is used to confirm the identity of VH sequences and to identify the associated VL.

Recently, by coupling a laser-based high-throughput colonies’ isolation with MiSeq Illumina sequencing, the Kwon’s lab set up a TrueRepertoire platform for the identification and retrieval of scFvs from a phage library, preserving the linkage information between heavy and light chains. Although thousands of single colonies can be easily analyzed, the repetitive identification of identical clones and the throughput of this platform enables the identification of hundreds of unique scFvs thus, placing this interesting technology in-between classical ELISA and NGS-guided screening [91].

## 4. Maximizing mAb Production for Massive Parallel Characterization

The growing demand for monoclonal antibodies to address an increasing spectrum of therapeutic indications emphasizes the need for speeding up the screening and the progress from candidate mAb identification into preclinical settings and then, if successful, into the clinical stage. Second and third-generation sequencing technologies have been successfully applied for monitoring antibody fragment libraries (i.e., phage, yeast, ribosome, etc.) [92]. More recently, single B cell screening has also taken advantage of NGS to explore the human B cell receptors’ landscape, thus accelerating the identification of monoclonal antibodies [93,94,95]. In the latter case, B cells are isolated from peripheral blood mononuclear cells (PBMC) of infected or vaccinated donors by flow cytometry. Following RT-PCR, V_H_ and V_L_ sequences are rescued and used to generate human mAbs.

Recognizing and eliminating the bottlenecks downstream of the screening is even more relevant for preliminary identification and testing of large numbers of lead candidate antibodies derived from a HTS. Indeed, the NGS-throughput coupled to effective rescue strategies allows to identify hundreds of potentially active antibodies or scFvs, which moved the major bottleneck of mAbs discovery from screening to the downstream processes of cloning and production.

Manufacturing of antibodies requires the cloning of the variable domains of scFvs into expression vectors to transduce host cells, which will thus generate the recombinant antibodies. A collection of vectors encoding a repertoire of constant domains can be used as a toolbox, to rapidly convert scFvs of interest in different antibody formats and isotypes (e.g., scFv-Fc, full antibody, bispecific Ab) (Figure 6) [96,97,98]. Classical cloning by ligation presents several constraints, limiting its scalability to tens of scFvs. Ligation-free technologies circumvent, at least in part, these limits. In-Fusion cloning allows to efficiently graft VH_s_ and VL_s_ into the appropriate vectors, encoding the constant heavy chain of the desired isotype (e.g., human IgG1, IgG4 or murine variants) as well as encoding for kappa or lambda light constant chains. The possibility of inserting any PCR fragment into any destination vector without the constraints of suitable restriction sites enables the rapid cloning and isotype switch with a success rate of over 95% [99]. Although the background is low, the use of a negative selection marker (e.g., toxic *ccdB*-gene) in the cloning region of the vectors further improves the confidence of cloning and makes the system ideal for high-throughput workflows (Figure 6A) [100].

The recombinant mAbs production step can be challenging, particularly on the small scale required for preliminary characterizations of mAb candidates. Although a range of expression systems have been developed (e.g., yeast, bacteria, insect cells), only mammalian cells fulfill the requirements for full post-translational modifications (i.e., glycosylation patterns) that can affect mAb’s immunogenicity, effector functions and stability [101]. While the clinical-stage antibodies still require stably expressing cell platforms yielding mAb titers of grams per liter, the process is time-consuming and expensive and thus not adequate for early-stage developments, particularly when the production of hundreds of mAb candidates is required [102].

To get cheap and fast mAbs production, transient expression processes are much more suitable, enabling rapid and parallel production of candidate mAbs (Figure 6B). This tool is thus ideal in the mAb discovery processes, to compare a hundred candidates from HTS in binding and biochemical assays, as well as to evaluate the therapeutic potential for selected preclinical candidates [103]. While transient production is fast and flexible, the mAbs’ yield rarely approaches hundreds of milligrams/Liters. In the effort to address this limitation, many technologies have been applied for improving the moderate yields and to scale up transient production in batch or fed-batch bioreactors to generate even grams of recombinant antibodies [104]. Among the most effective approaches, the improvement of cell viability, the enhancement in transcription and the optimization of the translation-folding-secretion axis have led to improved recombinant antibodies’ yields to hundreds of milligrams per liter of culture [105,106,107]. The most common mammalian cell line used for the production of recombinant proteins are Chinese hamster ovary (CHO) cells and baby hamster kidney (BHK21), however, these non-human cells can introduce post-translational modifications that are not present in human cells, thus requiring to select sub-clones with an acceptable glycan profile [108]. As an alternative to non-human cell lines, human HEK293 cells have been widely adopted for both the transient and stable production of recombinant proteins. The semi-stable production of recombinant proteins in HEK293 derivatives is noteworthy. HEK293 cell lines expressing the Epstein–Barr virus nuclear antigen 1 (EBNA1) (293E) or the SV40 large-T antigen (293T), allow for the episomal propagation of plasmids containing the matching origin of replication, thus enabling much higher yields, compared to the regular transient expression [99,109,110].

In the effort to enhance the translation of mRNAs encoding secreted recombinant proteins, we engineered the HEK293E cell line; with this aim, we exploited the technology of long non-coding RNA called SINEUP, whose activity depends on two elements: an inverted SINEB2 sequence able to enhance the recruitment of the translation machinery (effector domain, ED) and a 5′ domain overlapping to a target mRNA in a divergent head-to-head configuration (binding domain, BD) [111,112]. Since their discovery as natural antisense controlling Uchl1 translation [113], SINEUPs have attracted great applicative interest in improving the production of recombinant proteins [114,115,116,117], and to rescue defects associated with haploinsufficiencies [118,119].

Although the SINEUPs have been applied to boost yields of recombinant antibodies (and alternative antibody formats) in transient systems, the need to co-transfect the target-expressing plasmids and the SINEUP also limits the application of this technology [120,121]. To overcome this issue, we engineered HEK293E to stably express a SINEUP targeting the region encompassing the 5′UTR and signal peptide of heavy and light chain mRNAs with its binding domain (Figure 7) [122]. The combination of EBNA1 protein and SINEUP allowed HEK293EBNA_SINEUP to produce mAbs with proper, unaltered post-translational modifications in a semi-stable manner up to 300 mg/L. The targeting of signal peptide enables to fulfill boost in the production of any antibody without limitations of species (e.g., human, mouse), isotype (e.g., IgG1, IgG4) and formats (e.g., full mAb, scFv-Fc), offering a valuable opportunity to speed up the mAb discovery process (Figure 7). This strategy was also successfully implemented in CHO cells, bringing SINEUP technologies closer to the clinical manufacturing requirements [123].

## 5. Conclusions

Monoclonal antibodies have become the best-selling drugs in the pharmaceutical market. Their success relies on the versatility coupled to the high binding affinity and specificity for potentially any target of interest. More recently, antibodies have also been exploited as molecular shuttles to restrict the delivery of highly toxic agents. This is the case of immunocytokine and antibody-drug conjugates (ADC) where pro-inflammatory payloads (e.g., IL2, IL12) and chemical cytotoxic agents can be delivered to the tumor niche, leading to low systemic adverse effects and potentiating their anti-cancer activity [124,125,126,127]. Whatever function is amenable, the selection of high-affine and specific antibodies is a crucial step for the therapeutic success of mAbs. Display platforms, single B cell screening and hybridoma are currently the principal technologies for isolating mAbs. While the technological development for single B cell screening and hybridoma technologies have been recently reviewed [12,94,95,128], the novel technical improvements for display technologies were never systematically described. Each platform has its advantages and disadvantages; however, the scope of this review was to describe the most advanced tools related to display technologies that follow the entire workflow of the discovery process—from the screening to massive parallel biochemical characterization. Here, we summarize current knowledge in the rapidly progressing field of the NGS-guided high-throughput screening applicable to almost all display platforms (e.g., ribosome, phage, yeast), with an emphasis on limitations and future perspectives [129]. We also summarized the main differences between NGS-guided and classical ELISA screening of phage display libraries of scFvs in Table 1, where the advantages and disadvantages of both platforms are highlighted. Not long ago, many patents covering phage display technology limited its commercial exploitation. From 2010 to 2021, most of the key patents have expired, encouraging academic groups and biotech companies to accelerate the implementation of this technology for the discovery of fully human antibodies [36,130]. The strength of display technologies is related to their versatility and to the possibility of targeting any antigen of interest, including toxic and poorly immunogenic molecules. Moreover, non-canonical antibody formats (i.e., scFv, single domain, nanobody), that are emerging as a useful tool to develop CAR-T cells, bispecific and other biopharmaceuticals, can be isolated by phage display technology, overcoming the challenge of antibody reformatting. Despite this feature being able to help in the isolation of non-canonical formats, it also could represent a limitation during the scFv to mAb reformatting, as some scFvs have been found to lose affinity, stability or folding during conversion into IgGs [33,131]. The ScFv-Fc format circumvents at least, in part, this issue, as it mimics the bivalent binding and effector functions of full mAb, still preserving the scFv layout [132]. An additional limitation of phage display has been for years, the covering of all antigen-specific mAbs present in a given antibody library. Indeed, even if the panning cycles allow for pulling out potential binders, after three or four selection rounds, few clones tend to take over the rare antigen-binding ones [133]. Standard ELISA coupled to phage display often results in the repetitive isolation of these few dominant clones, losing information on rare binders. Finally, display technologies suffer from the accumulation of false-positive clones, attributable to both non-valid inserts and target-unrelated clones. Typically, the clones bearing non-valid inserts encode stop codons and indels inhibiting the expression of functional coat protein-antibody fusions. The absence of fusion proteins permits the integration of coat protein from helper phage, allowing their amplification in *E. coli* as selection-unrelated contaminants. On the other hand, the enrichment of selection-related, target-unrelated clones that bind to screening components beyond the target itself (e.g., plastic, Fc fragment, biotin, BSA, etc.), results in the dilution of potential binders. Many studies are continually developing methods to overcome these limitations and for high-throughput screening of display libraries. The link between the genotype (scFv coding sequence) and the phenotype (surface-displayed scFv) is the principle for the implementation of sequencing-guided screening of display libraries. Next-generation sequencing allows high-throughput interrogation of the selection steps to evaluate the relative enrichment of scFvs as indicative of their affinity for the target of interest, giving a far greater coverage compared to classical ELISA screening. This coverage also allows the identification of rare clones that, in some cases, can target sub-dominant epitopes with intriguing properties. Beyond the enrichment, high-throughput sequencing allows to discard a priori all those clones bearing non-valid inserts to restrict the analysis to potential binders. To further improve the confidence of binding prediction, we are implementing a database collecting predicted and validated binders to target-unrelated components of the biopanning. Refining the sequencing-guided screening, by applying these informatic scripts, grants dedicated wet step efforts to those scFvs with high-potential to be specific binders. Moreover, the characterization of these false-positive clones’ sequences (target-unrelated and selection-unrelated), could lead to an increased understanding of how these clones arise to intervene in crucial selection steps for preventing or limiting this phenomenon. Although next-generation sequencing improved the quality of phage display, there are at least two key limitations: (1) the maximal read length of second-generation sequencing technologies (i.e., Illumina) allows to sequence only a portion of scFvs (e.g., VH), making the reconstruction of the entire antibody fragment challenging; (2) to physically fish out in silico-identified clones of interest, a laborious recovery step is necessary. Long reads obtainable from third-generation sequencing allow achieving quali-quantitative information of the entire scFvs. Not long ago, the relatively low-throughput, the high error rate, and the cost limited the exploitation of third-generation sequencing in display technologies. The circular consensus reads that the more widespread platforms’ availability is making this paradigm shift overcome challenges with the higher sequencing error rate, and also leads to a reduction in sequencing costs [65]. The promise of the application of third-generation sequencing is emerging in recent literature, demonstrating that PacBio circumvents the issue of VH and VL matching [66,67,68]. Furthermore, besides preserving the linking information of VH and VL, the obtaining of full-length scFvs’ sequences could, in the future, also circumvent the need to physically retrieve phages’ clones from the sub-library by synthesizing the VH and VL of interest, thanks to the progressively lowering of synthetic DNA cost services (<0.1 US dollar/base) [134]. This perspective, alongside advances in high-throughput reformatting of phage displayed antibody fragments to IgGs [99,135,136], as well as to the implementation of engineered mammalian cell factories for improved production [122,137], will soon make it possible to efficiently test antibodies from display libraries in the end-use format, facilitating their preclinical development. Recently, also machine learning and neural networks have been implemented to optimize the screening itself and antibodies’ properties [138,139].

Furthermore, the future challenge to widespread mAbs’ use will likely be related to manufacturing costs, as this latter is still a major hurdle associated with mAbs. The high cost per patient is especially attributable to both high doses and the continuous administrations, especially needed in the two fastest-growing therapeutic areas of chronic diseases and cancer immunotherapy [130,140]. One of the remarkable innovative approaches is the in vivo gene transfer of mAbs. This technology—also currently investigated in clinical trials—is particularly attractive for the management of chronic diseases [141]. Indeed, as an alternative to the administration of mAb proteins, gene transfer resolves to administer the genetic information for mAb, allowing the patient to produce therapeutic mAbs for a sustained period, and thus avoiding the repetitive passive administration of recombinant proteins. The recent achievements in genetic medicine reported for COVID-19 vaccines (viral vectors and mRNA) could definitively validate this approach, opening a new era of antibody success [142,143].

## Figures and Tables

**Figure 1 cancers-14-01325-f001:**
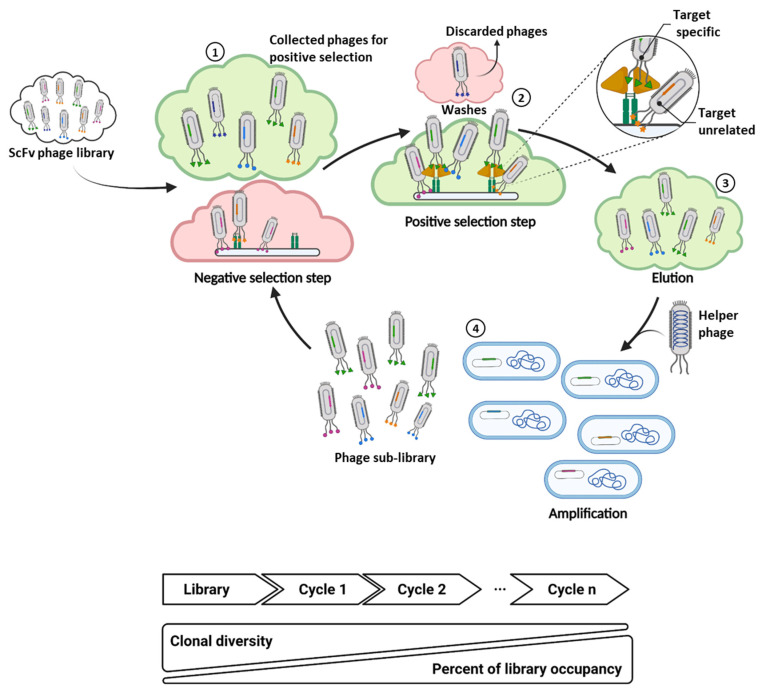
The cartoon depicts the entire procedure of in vitro biopanning. The biopanning of phage display libraries involves three principal steps for selection. **①** A negative selection is performed by incubating the scFv library (and sub-libraries) with all the components of the screening (in the example Fc-Tag and solid-phase) except for the antigen of interest to discard target-unrelated clones. In the positive selection step, phages are incubated with the antigen of interest **②**; unbound phages are washed out by extensive washes. Residual target-unrelated clones are indicated in the zoomed circle as a binder to Fc-Tag. Bound phages from positive selection are eluted **③** and are amplified in *E. coli* by co-infection with helper phage **④**. All these steps are repeated several times to enrich the target-specific clones.

**Figure 2 cancers-14-01325-f002:**
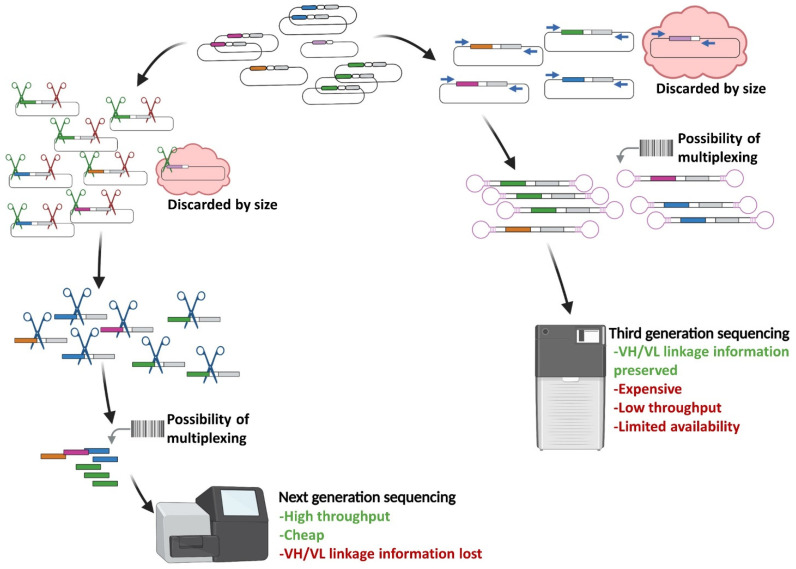
Strategies to analyze phage library by second- and third-generation sequencing. On the left, the workflow for extraction of variable fragments compatible with second-generation sequencing platforms (e.g., Illumina MiSeq) is shown. Full-length scFvs and VH fragments are extracted from dsDNA of phages from *E. coli*. On the right, the workflow for the application of third-generation sequencing of entire scFvs by circular *consensus* sequencing is represented. Molecular barcoding can be included in both strategies for multiplexing samples. The indicated pros (green font) and cons (red font) are referred to as the current state-of-the-art.

**Figure 3 cancers-14-01325-f003:**
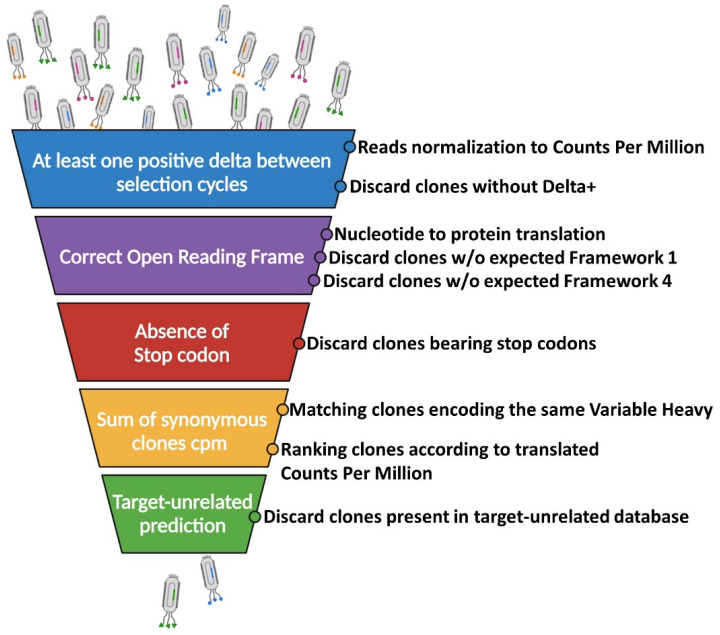
Funnel pipeline applied to sequencing data to improve the confidence of in silico binding predictions.

**Figure 4 cancers-14-01325-f004:**
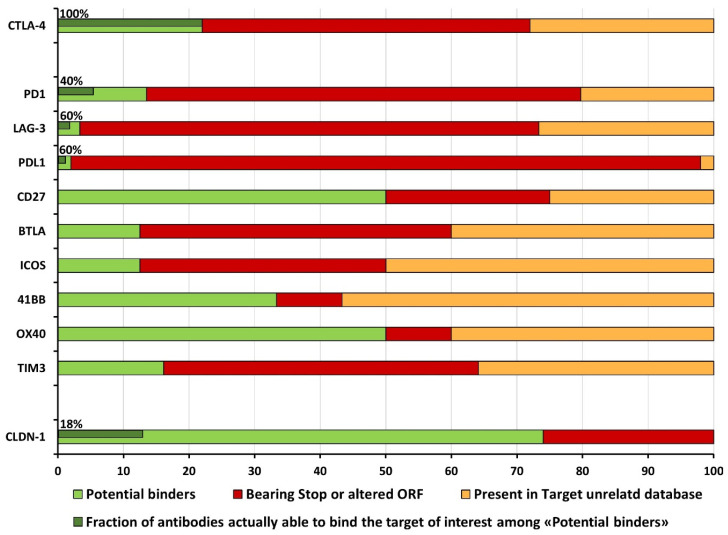
Snapshot of sub-libraries composition in different scFv phage display screenings. Potential binders, clones bearing stop codons or altered ORFs, and target-unrelated clones are respectively shown in light green, red and orange. Dark green indicates the percentage of specific binders validated by ELISA assays, where available. Ref relative to the screening: CTLA-4 [47]; PD1 to TIM3 [33]; CLDN-1 [61].

**Figure 5 cancers-14-01325-f005:**
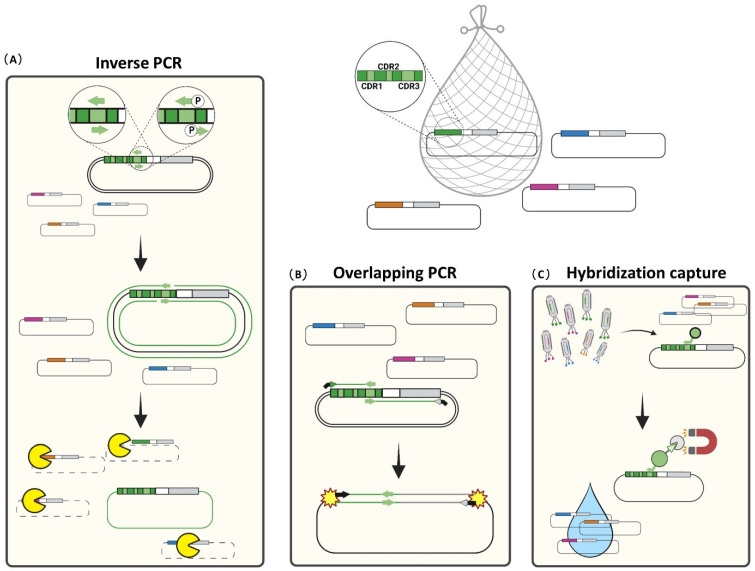
Rescue strategies to physically isolate a hypothetical clone of interest (green) from a given sub-library. (**A**) In the inverse PCR approach, two primers are designed into the HCDR3 of the scFv to obtain the newly synthesized, entire phagemid vector of interest. *Dpn*I enzyme is used to cleave and remove template DNA containing methylated and hemi-methylated restriction sites (G^m^A|TC) preserving the newly synthesized, unmethylated DNA. (**B**) In overlapping PCR, primers into the HCDR3 region are combined with primers designed into constant HFR1 and LFR4. The two amplicons are assembled by overlapping PCR and cloned into an empty phagemid vector. (**C**) In hybridization capture, functionalized primers specific to HCDR3 are used as bait to isolate the clone of interest.

**Figure 6 cancers-14-01325-f006:**
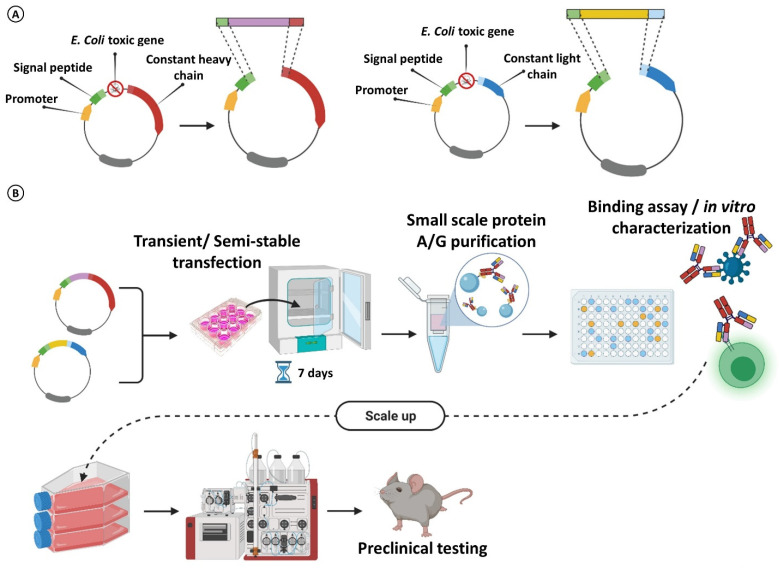
Workflow for scFvs conversion into full mAbs and small/medium scale production. (**A**) Variable heavy and light chains of an scFv can be subcloned into mammalian expression vectors encoding constant domains of antibodies. A toxic gene can be used to improve the confidence of cloning. Any format or isotype of interest can be produced (full mAb, scFv-Fc, etc.). (**B**) Small scale (e.g., 12-well plates) transient/semi-stable transfections in the producer cell lines allow for obtaining enough antibodies for preliminary in vitro testing. The transfection/purification process can be scaled up for preclinical testing.

**Figure 7 cancers-14-01325-f007:**
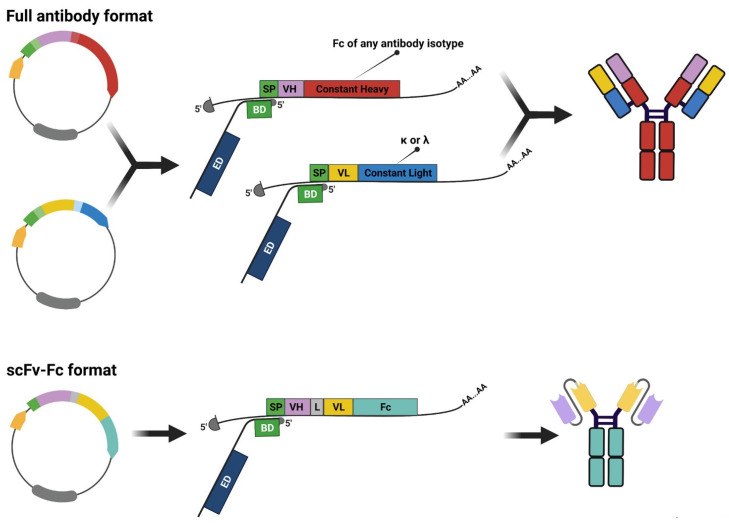
Long non-coding SINEUP RNAs can be used to improve the production of recombinant antibodies (full mAbs or scFv-Fc) by divergent 5′ end head-to-head complementarity with the target coding mRNA. The binding domain of SINEUP has been designed to target the region encompassing 5′UTR and signal peptide of any secreted recombinant protein.

**Table 1 cancers-14-01325-t001:** Comparison between NGS-guided vs. classical ELISA screening. Advantages and disadvantages are respectively indicated as green and red.

	NGS-Guided	Classical (ELISA)
Yield of positive hits (Number of different target-specific scFvs)	Hundreds (Top enriched and rare clones)	Few up to tens (Only top enriched)
Specificity confidence (Fraction of scFvs confirmed as specific by biochemical assay)	High	Low
Handling	Rescue step needed to physically isolate clones	Repetitive ELISA assay
Sequence information; VH to VL matching	- Short reads limits sequence information to VH; VH/VL matching lost (Illumina)	Entire scFv sequence available after picking (Sanger)
	- Entire scFv sequence and VH/VL matching available (Third-generation)	

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
