# Peer review of "High-Throughput Monoclonal Antibody Discovery from Phage Libraries: Challenging the Current Preclinical Pipeline to Keep the Pace with the Increasing mAb Demand"

_cancers, 2022, doi:10.3390/cancers14051325_

Round 1
Reviewer 1 Report
This review article by Zambrano et. al. has summarized novel phase display technologies for high-throughput monoclonal antibody discovery from phage libraries. This is a very well-written review article citing relevant literature and discussing recent advances, the pros and cons of each technique. Overall, this review article will help researchers from diverse academic and industry backgrounds to understand new advances in strategies for the identification of mAbs using phage-display libraries.
Author Response
We really thanks the reviewer for appreciating our work.
Reviewer 2 Report
The authors wrote a timely review on the topic of NGS-guided antibody discovery from phage display libraries. The authors elegantly presented antibody discovery approaches from the in vitro biopanning to advanced methods incorporating the NGS methods. Importantly, a funnel pipeline as applied to phage libraries with NGS-based strategies to better select clones with in silico predictions as well as discarding clones appear in target-unrelated database is very useful, for example, to the antibody drug discovery community. It is also highlighted that the NGS-method also enable the clone rescuing strategies.
Minor points and comments/suggestions:
- In Figure 3, for CPM and delta+ (also VH, FR) expand the abbreviations.
- Figure 4, legend' last word (publications) is misplaced at line 448.
- In line 148, do the authors mean "large repertoires" for "large antibody repertoire"? if so, add antibody .
- Phage display played a role in the discovery of Adalimumab discovered but it should be noted that the process known as “guided selection” using a murine mAb as the original template could be the key.
- Figure 4 and the data associated are very informative. Is it possible to have any rare clones (any evidence from authors/literature) present in Target unrelated database also bind to the target? Is there any affinity data to the target for those clones (found in Target unrelated)?
- Any formatting preference for scFv vs Fab to phage or yeast?
- Authors can mentioned about the recent trends about phage libraries with tailor-made properties (ex. developability) and machine learning approaches alongside antibody libraries.
- The possibility of NGS-based library method for discovering rare clones may be elaborated.
- The review can be shortened and focused on the key topic/message.
- Conclusion has some parts that sound like introduction' materials or repeating. An effective, shorter conclusion will be better.
Author Response
We really thanks the reviewer for appreciating our review.
Below a point by point response to minor revision:
- In Figure 3, for CPM and delta+ (also VH, FR) expand the abbreviations.
- Figure 4, legend' last word (publications) is misplaced at line 448.
- In line 148, do the authors mean "large repertoires" for "large antibody repertoire"? if so, add antibody .
- Phage display played a role in the discovery of Adalimumab discovered but it should be noted that the process known as “guided selection” using a murine mAb as the original template could be the key.
We corrected these points (1-4) - Figure 4 and the data associated are very informative. Is it possible to have any rare clones (any evidence from authors/literature) present in Target unrelated database also bind to the target? Is there any affinity data to the target for those clones (found in Target unrelated)?
This is a very interesting point. To enter in target-unrelated database, a sequence need to be present at least in two screenings. - Any formatting preference for scFv vs Fab to phage or yeast?
To improve display we often prefer scfv format over Fab. - Authors can mentioned about the recent trends about phage libraries with tailor-made properties (ex. developability) and machine learning approaches alongside antibody libraries.
We included this point in discussion - The possibility of NGS-based library method for discovering rare clones may be elaborated.
We added a sentence about rare clones. - The review can be shortened and focused on the key topic/message.
- Conclusion has some parts that sound like introduction' materials or repeating. An effective, shorter conclusion will be better.
We shortened conclusion
Reviewer 3 Report
This manuscript is a very good job going through the development of mAb screening from phage display library, NGS and in silicon bioinformatics.
I just wonder that authors might contribute a comparison table showing pros&Cons Among the throughput, timing, yields, false-positive candidates, commercialized applications and clinical translation from the development of mAB screening of traditional ELISA to most updating intercrossing technologies.
Author Response
We really thanks the reviewer for appreciating our manuscript. We included "table1" to compare NGS-guided vs Classical ELISA
Round 2
Reviewer 2 Report
I satisfied with the revised version.
In line 699, is the usage of "latter" correct? or "later". I'd suggest the following:
"That said, the future challenge to widespread mAbs’ use will be likely related to manufacturing cost as this latter is still a major hurdle associated with mAbs." =>
"Finally, the future challenge to widespread mAbs’ use will be likely related to manufacturing cost which still remains a major hurdle associated with mAbs."